# Towards Novel Biomimetic *In Vitro* Models of the Blood–Brain Barrier for Drug Permeability Evaluation

**DOI:** 10.3390/bioengineering10050572

**Published:** 2023-05-10

**Authors:** Inés Mármol, Sara Abizanda-Campo, Jose M. Ayuso, Ignacio Ochoa, Sara Oliván

**Affiliations:** 1Tissue Microenvironment (TME) Lab, Aragón Institute of Engineering Research (I3A), University of Zaragoza, 50018 Zaragoza, Spainsabizanda@dermatology.wisc.edu (S.A.-C.); 2Institute for Health Research Aragón (IIS Aragón), 50009 Zaragoza, Spain; 3Department of Dermatology, Department of Biomedical Engineering, and Carbone Cancer Center, University of Wisconsin-Madison, Madison, WI 53705, USA; jayuso@dermatology.wisc.edu; 4CIBER Bioingeniería, Biomateriales y Nanomedicina, Instituto de Salud Carlos III, 28029 Madrid, Spain

**Keywords:** blood–brain barrier, organ-on-chip, *in vitro* models, microfluidics, drug screening

## Abstract

Current available animal and *in vitro* cell-based models for studying brain-related pathologies and drug evaluation face several limitations since they are unable to reproduce the unique architecture and physiology of the human blood–brain barrier. Because of that, promising preclinical drug candidates often fail in clinical trials due to their inability to penetrate the blood–brain barrier (BBB). Therefore, novel models that allow us to successfully predict drug permeability through the BBB would accelerate the implementation of much-needed therapies for glioblastoma, Alzheimer’s disease, and further disorders. In line with this, organ-on-chip models of the BBB are an interesting alternative to traditional models. These microfluidic models provide the necessary support to recreate the architecture of the BBB and mimic the fluidic conditions of the cerebral microvasculature. Herein, the most recent advances in organ-on-chip models for the BBB are reviewed, focusing on their potential to provide robust and reliable data regarding drug candidate ability to reach the brain parenchyma. We point out recent achievements and challenges to overcome in order to advance in more biomimetic *in vitro* experimental models based on OOO technology. The minimum requirements that should be met to be considered biomimetic (cellular types, fluid flow, and tissular architecture), and consequently, a solid alternative to *in vitro* traditional models or animals.

## 1. Blood–Brain Barrier

The structure of the brain’s microvasculature is named the blood–brain barrier (BBB). The BBB is a complex, highly selective barrier that ensures the oxygen and nutrient supply while avoiding the entry of pathogens and potentially harmful molecules. The main goal of the BBB consists of keeping the brain separated from the rest of the organism and providing an adequate environment for the delicate brain cells [1]. Alterations in BBB integrity and function are related to the central nervous system (CNS)-related disorders, such as Alzheimer’s disease, Parkinson’s disease, or amyotrophic lateral sclerosis [2,3,4]. Recently, BBB dysfunction was also described in COVID-19 patients [5,6]. However, in this pathological context, the presence of the BBB implies an issue for drug delivery for the management of those conditions [1].

The BBB structure includes a cellular component as well as the extracellular matrix scaffolding. The cellular component of the BBB, the so-called neurovascular unit (NVU), is composed of five different cell types, namely endothelial cells, pericytes, astrocytes, microglia, and neurons (Figure 1; Table 1). On the other hand, the non-cellular component is the basal membrane or the basal lamina (BL), which provides structural support to the NVU and influences cell behavior.

Endothelial cells from cerebral microvasculature are, to a great extent, responsible for the highly selective permeability of the BBB. The paracellular trespassing of hydrophobic molecules is limited due to the establishment of tight junction between neighboring endothelial cells. Tight junctions are mainly composed by claudin-5 and occludin. Endothelial cells further express different substrate-specific transporters to allow the entry of essential molecules from the bloodstream to the brain parenchyma, as well as the release of waste products. Moreover, endothelial cells produce and secrete most of the proteins that compose the basal lamina to which they are attached [1,7].

Pericytes are a heterogeneous group of mural cells from the circulatory system that covers the endothelial surface. One of the main functions of these cells consists of ensuring the correct functioning of endothelial cells. Pericytes secrete angiopoietin, which induces occluding expression, thus contributing to the correct maintenance of tight junctions. Moreover, pericytes secrete further substances, such as transforming growth factor (TFG-β), that influence endothelial function. Consequently, defects in pericytes lead to a dysfunction in BBB permeability, which is increased. Pericytes are also involved in the regulation of capillaries’ blood flow, contribution to the formation and maintenance of basal lamina, and regulation of vesicle endothelial transcytosis and immune cell infiltration [1,7,8].

Astrocytes play multiple functions, including nutrient metabolism, inflammatory response, and tissue repair. In the context of the BBB, the astrocyte end feet cover almost the whole basolateral surface of endothelial cells, connecting them with neurons. As pericytes, astrocytes secrete molecules that influence endothelial cell behavior, such as TFG-β. Astrocytes also regulate the entry of immune cells into brain parenchyma to avoid neuropathological inflammatory responses [1,7,9].

Microglia are part of the neuroimmune system, separated from the systemic immune system by the BBB, and responsible of responding to insults such as pathogens. Microglia are involved in phagocytosis, antigen presentation, and the activation of inflammatory response. Microglia might adopt different phenotypes, such as M1 and M2, which play pro-inflammatory and anti-inflammatory roles, respectively [1,7,10]. Moreover, microglia are involved in the maintenance of BBB integrity due to its close connection with pericytes. Therefore, microglia might be also involved in blood flow regulation, despite further analysis being needed [8].

Despite the great relevance of neurons for organism function, the role of neurons for NVU integrity remains poorly studied. Unlike the rest of the cell types that integrate the NVU, neurons do not provide a physical barrier to avoid substance entry to brain parenchyma. Instead, some data support that NVU neurons might promote changes in vessel tone in response to brain parenchyma metabolic demands by releasing factors such as nitric oxide. This capacity might be mediated by the connection between NVU neurons and the rest of the brain neurons, along with the great sensitivity of this cell type to fluctuations in oxygen and further nutrient levels [11]. Those factors, which modulate the functioning of the other cell types, consequently trigger vasoconstriction and/or vasodilatation according to brain needs. Astrocytes are especially well connected with neurons, and some research suggest that neuronal signals might modify blood flow throughout astrocytes mediation. However, since neurons are in direct contact with blood vessels, it is unclear whether those cells are able to directly trigger blood flow modifications [1,7]. Some authors suggest that NVU neurons are involved in cerebral angiogenesis during development. However, astrocytes-mediated regulation might become of greater relevance to cerebral vasculature flow upon post-development [12].

Finally, in relation to the cell architecture of the BBB, the presence of the BL stands out. BL is the extracellular matrix that provides structural support to the cells that integrate the BBB. The cells that conform to the NVU, mainly endothelial cells, pericytes, and astrocytes, synthesize the proteins that integrate BL [1,13,14]. Endothelial cells attach BL through integrins, which are surface receptors, and this interaction results in changes in cell behavior, including proliferation and survival. Therefore, this structure is involved in cell organization as well as in BBB integrity maintenance. BL is composed mainly by type IV collagen, fibronectin, heparan sulphate proteoglycan, and laminin. Different isoforms of laminin were identified, and the balance between them is relevant for BBB formation. Interestingly, each cell from the NVU synthesizes different isoforms of laminin, resulting in a differential distribution pattern on BL [15].

## 2. Challenges in Drug Delivery to the CNS

Despite protecting the delicate brain parenchyma from external insults, the presence of the BBB is a great limiting factor in terms of drug delivery. The entry of macromolecular drugs, including monoclonal antibodies and peptides, is completely prevented, and only a low percentage of small molecule drugs is permeable through the BBB [16,17]. 

Molecules can access brain tissue mainly through two different pathways: paracellular transport, namely between the space of neighboring endothelial cells; or transcellular transport, through the endothelial cell (Figure 1) [16,17]. Paracellular transport is mediated by channels that allow size-dependent diffusion of molecules [18]. Ions and other small, water-soluble molecules are able to enter through the paracellular pathway thanks to concentration gradients. Small hydrophilic molecules might also enter using aqueous channels. However, molecules greater than 4 nm are unable to penetrate through this via [17]. The main challenges for drug delivery into the brain through the paracellular pathway are tight junctions between endothelial cells, which display unusually high electrical resistance [18,19].

Some research focused on achieving a temporary and reversible disruption of BBB integrity to enhance the paracellular transport of drugs. Hyperosmotic agents, e.g., mannitol and glycerol, trigger high osmotic pressure that leads to a shrinking of endothelial cells, and consequently, to a disruption of tight junctions that allows for drug diffusion into brain tissue. Other alternatives were explored. Further approaches include vasoactive compounds, a combination of focused ultrasound and microbubbles, or local hyperthermia induced by magnetic nanoparticles exposed to a magnetic field, among others. However, most of these methods are highly invasive and might allow the penetration of potentially harmful substances [20,21]. Therefore, less dangerous strategies might be adopted. Non-invasive strategies are mainly focused on drug modifying, e.g., enhancing its lipid solubility to improve BBB penetration [21].

On the other hand, transcellular transport faces further issues. Transcellular migration requires molecule uptake by endocytosis and exocytosis to release them on the other side of the BBB [22]. Small nutrients and metabolites (e.g., glucose and vitamins) and other relevant molecules, such as insulin, transferrin, or low-density lipoproteins (LDL) particles, might reach the brain parenchyma through this route. However, the presence of specific receptors and substrate carriers strongly limits the amount of molecules able to enter through the transcellular pathway [19]. Small lipophilic compounds that penetrate endothelial cells through passive diffusion must face the battery of enzymes related to drug metabolism, such as cytochrome P450, which are highly expressed on the endothelial cells of the BBB. Moreover, those cells overexpress ATP-binding cassette transporters (P-glycoprotein, BCRP, and MRP2) to actively promote drug release from the cell. Other cells of the NVU also display similar transporters to pump out drugs from the cell [16,23]. These unique features convert the BBB into an extraordinarily effective barrier. Some authors proposed that pharmacological inhibition of such efflux transporters might enhance intracerebral drug delivery [21].

Interestingly, despite the fact that endothelial cells are directly responsible for molecule permeability through the transcellular pathway, these cells are not the only ones involved. The close relation among endothelial cells, pericytes, and astrocytes strongly influences the fate of the transcellular pathway [22]. Furthermore, results from Kutuzov et al. suggest that the glycocalyx that coats the luminal side of endothelial cells might act as a first barrier [24]. Glycocalyx avoids the entry of large molecules from blood to brain parenchyma, although small molecules are able to cross it through. In order to develop a realistic *in vitro* model of the BBB for drug delivery studies, every component involved in permeability should be carefully considered and included.

Drug and drug delivery system designing for the management of brain-related disorders must, consequently, carefully consider every parameter that ensures a successful penetration into the BBB: size, molecular weight, solubility, surface charge distribution, etc. [17]. In this line, smart drug delivery systems might offer several advantages in comparison to traditional drug delivery systems. Further details are included in the extensive review from Lynch and Gobbo, in which the advantages of nanoparticle-based smart drug delivery systems crossing through the BBB versus traditional drug delivery systems are deeply discussed [25]. Briefly, ligand decoration is crucial to ensure an optimum transcellular transport of the nanomedicine, especially to promote receptor-mediated transcytosis, which is the most used method for intracerebral delivery. The most common receptors that lead the surface functionalization process are those over-expressed in endothelial cells from brain microvasculature, such as the transferrin receptor, LDL receptor, and insulin receptor, among others [20].

However, Lynch and Gobbo also pointed out the issues for clinic translation, mainly related to the lack of realistic models [25]. Intracerebral drug delivery then faces two great challenges. On one hand, drug and/or drug delivery system design must ensure BBB penetration. On the other hand, biomimetic models are needed to guarantee accurate, reliable preclinical data.

## 3. Towards Biomimetic Models of the BBB

Historically, animal models were widely used for the development of novel drugs for the management of brain-related disorders. Most animals display a barrier that separates the brain from the bloodstream [26]. However, the existence of interspecies differences might display a great impact on the obtained results. Those differences might be substantial, such as the cellular composition of the barrier, which differs among species, or more subtle [26]. In this context, although the amino acid sequence of the efflux transporter P-gp is highly conserved among species, differences in substrate recognition, expression, and efficacy can be observed [27]. This is of great relevance when analyzing the potential toxicity of a drug candidate, since P-gp expression in mouse and rat is higher than in human. As a consequence, the toxicity of a drug might be under- or overestimated [28]. Taken together, those and further differences between species result in a high percentage of failure when translating promising drug candidates from animal models to clinical trials. In addition, working with animal models implies high economic costs and ethical issues.

Therefore, a great effort was put into developing novel robust, biomimetic *in vitro* cell-based models. First, models were focused on reproducing tight junctions between endothelial cells, which were grown on traditional plastic culture dishes [29,30]. Such 2D and static models displayed several limitations. Firstly, it is well documented that some features of endothelial cells from the cerebral microvasculature are strongly influenced by shear stress [31,32]. Furthermore, as aforementioned, other cells from the NVU are closely connected to endothelial cells and influence their function [1,7,8,9]. Models that ignore this connection are unable to mimic *in vivo* conditions. Therefore, trends in BBB *in vitro* models moved towards co-culture in order to include different cell types. Moreover, traditional 2D static models used to be based on animal cells rather than human cells. Given the previously mentioned differences between human and animal BBB, those models were unable to reliably reproduce the unique architecture of the human barrier [29,30]. In order to overcome those limitations, animal-derived cells were gradually replaced by human cells and novel models arose as potential substitutes of culture dishes.

Transwell-based models were widely used to generate *in vitro* BBB models that include more than one cell type. This device contains two chambers, apical and basolateral, separated by a porous membrane. Endothelial cells are seeded in the apical side of the membrane, while further cells from the NVU can be added to the basolateral chamber [33] (Figure 2). In line with this, Stone et al. developed different Transwell models including two, three, or four different cell types, namely endothelial cells, astrocytes, pericytes, and neurons [34]. Authors noticed an increase in transepithelial resistance (TEER) as they increased the cellular complexity of the *in vitro* model. Since TEER is a measurement of electrical resistance across both chambers, this parameter is highly correlated with barrier integrity [35]. Thus, an increased TEER value might be interpreted as having higher barrier integrity. They concluded that this model mimics better *in vivo* conditions; however, this Transwell-based model does not incorporate flow, and consequently, endothelial cells are not exposed to shear stress. In this sense, Transwell models of the BBB represent a step toward mimicking human *in vivo* BBB, although still display several limitations.

Subsequently, some researchers focused on improving Transwell-based *in vitro* BBB models to expose endothelial cells to shear stress. Harding et al. generated a microfluidic device compatible with a Transwell model containing endothelial cells, pericytes, and astrocytes [36]. This kind of model overcame one of the limitations of Transwell models, although the BL was not included [29]. As previously discussed, the non-cellular component of the BBB is also involved in barrier integrity [15]. Hence, BL must be mimicked as well to ensure that the *in vitro* model reproduces *in vivo* conditions. Research from Katt et al. showed that a collagen gel might be added to the traditional porous Transwell membrane to imitate *in vivo* BL [37]. Authors observed significant changes in TEER according to modifications in gel composition, thus confirming that BL must be included in a realistic BBB model.

Although some of the unique features of cerebral microvasculature can be successfully reproduced in Trasnwell-based models, some key parameters, such as the *in vivo* cylindrical geometry, cannot be included. Whereas cylindrical geometry might not be directly involved in tight junction formation, and as a result, in barrier permeability, this parameter influences other features, such as morphology [38]. Therefore, cylindrical geometry might display an indirect role in endothelial cell function, and consequently, in that of the whole BBB [29,39].

In conclusion, the traditional *in vitro* models of the BBB present relevant limitations that need to be determined (Figure 3). Due to them, the development of *in vitro* BBB models moved towards different devices that get closer to the complex and dynamic microenvironment of cerebral microvasculature: microfluidic organ-on-chip devices.

## 4. Advantages of Organ-on-Chip Devices versus Traditional Cell Culture Systems

Organ-on-chip (OOC) technology was born due to the need to develop better models to reproduce human physiology and as an alternative to animal experimentation. This technology is based on the use of microfluidic devices, which offer superior cell and tissue manipulation due to the use of advances in microfabrication techniques and the predictable nature of fluid physics at the microscale (e.g., laminar vs turbulent flow) [40]. Due to this, OOC allows for a more reliable understanding of physiological function and reproducing pathological conditions [41,42,43,44,45,46].

One of the greatest advantages of the OOC devices is that they allow the co-culture of different cell types in three dimensions, mimicking living tissue microarchitecture. Growing in 3D is required for the development of some cellular and tissue features that might strongly influence drug response, including morphology and metabolic profile, which were reported to change in 2D culture in regard to 3D culture [47]. Furthermore, cell–cell interactions and cell–extracellular matrix interactions are closer to the physiological microenvironment. This allows for advanced cellular biology studies, e.g., the evaluation of immune cells migration and their cross-talk with cancer cells or modeling bacterial infection in the intestinal epithelium [48,49].

OOC also enables dynamic cell culture due to the incorporation of microfluidic channels that reproduce fluid or air flow. Therefore, environmental factors that cells might require for correct development, such as shear stress, are included in the model [45,50,51]. Finally, coupling biosensors might allow a continuous, real-time monitoring of different parameters of interest. Thus, Nashimoto et al. developed an electrochemical sensor able to detect changes in oxygen metabolism, which was coupled with a microfluidic device containing human lung fibroblast spheroids and patient-derived cancer organoids [52]. Zoio et al. were able to perform an on-chip measurement of TEER by developing a model with electrodes integrated to evaluate the barrier function of an *in vitro* skin model. In this sense, the incorporation of biosensors provides robustness and reliability to the *in vitro* model [53].

Therefore, OOC technology rose as a promising tool for the development of more advanced *in vitro* models that reproduce the complex structure of the human BBB [54]. Given that more than one cell type can be cultured simultaneously, all the cell types that integrate the NVU and influence barrier properties can be included. Furthermore, endothelial cells can be exposed to shear stress and the presence of the basal lamina can be mimicked. Thereby, BBB-on-chip might overcome some of the previously discussed limitations of static models and Transwell-based models. 

Briefly, biomimetic models of the BBB are based on microfluidic devices often compartmentalized by a membrane, mimicking the BL, which separates the model into two different areas: the cerebral vasculature and the brain (Figure 2). For the vascular area, where the endothelial cells are located, the device will have a channel to work under flow conditions and it will be the drug administration route. The pattern of the channel should allow the establishment of a tight junction between neighboring endothelial cells, resulting in the development of a functional endothelium. In direct contact with the membrane, pericytes and astrocytes will be seeded to establish the co-culture of the three cell types that make up the NVU and the BBB architecture. If the membrane is to be avoided in the microdevice, it will be necessary to include hydrogel to support the brain parenchyma, as described in the following section. In both cases, direct cellular contact between endothelial cells, pericytes, and astrocytes will be required for NVU integrity. In the next sections, different proposals of BBB-on-chip models will be discussed to deepen this concept and clarify some key points of biomimetic *in vitro* models.

Furthermore, biosensors might be coupled to the microfluidic chip for monitoring relevant parameters. One of the most analyzed is transendothelial electrical resistance, since, as previously mentioned, it might be extrapolated as a measure of barrier integrity. In line with this, Jeong et al. and Tu et al. are two examples of microfluidic BBB models that include systems for real-time monitoring of barrier integrity by measuring TEER [55,56].

However, Vigh et al. and Mir et al. pointed out that TEER measurement is highly influenced by biological—e.g., cell types included in the BBB model, cell passages, and physical—e.g., temperature and physicochemical parameters—e.g., cell medium composition and viscosity, as well as the setup measurement, namely electrode material, shape and position relative to cell monolayer, and the ratio between the electrode and membrane area [35,57]. Therefore, TEER measurement provides limited information regarding barrier integrity and further analysis is needed, such as permeability for paracellular markers [35]. Regarding the potential of biosensors in BBB-on-chip models, Mir et al. discussed the possibility of coupling those allowing the continuous monitoring of biomarkers relevant to neurodegenerative diseases, which might be of great utility for investigating drug effectivity or drug-induced neurotoxicity *in vitro* [57]. This approach might be adapted for analyzing biomarkers from pathologies other than neurodegenerative diseases, thus leading to the development of more relevant *in vitro* BBB microfluidic models.

## 5. BBB-on-Chip: Mimicking Tissue Microarchitecture

In order to reproduce the BBB, OOC microfluidic devices take the core idea of Transwell devices, separating two chambers with a membrane, and improve it to display certain advantages *versus* traditional models. Therefore, these devices make it possible to establish a ‘vascular’ side, on which endothelial cells are seeded, and a ‘cerebral’ side, where the rest of the cell types that integrate the NVU are grown. Regarding the advantages of microfluidic in comparison to static models, Santa-Maria et al. found that fluid flow triggered an upregulation of glycocalyx core protein genes, and consequently, an increase in the negative surface charge of the BBB [58]. Furthermore, authors reported that co-culturing pericytes and endothelial cells resulted in a higher decrease in surface charge, thus pointing out the relevance of including more than one cell type to enhance the biomimetic capacity of a microfluidic model. Herland et al. compared the permeability of the barrier developed on a Transwell device and on a microfluidic chip and observed that permeability was significantly higher on the Transwell device, thus suggesting that the BBB model constituted on the chip was closer to *in vivo* [59].

### 5.1. Porous Membranes

To develop a BBB-on-chip model and compartmentalize the cell culture in the model, different approaches can be followed. Most use a semipermeable porous membrane based on inert materials such as polycarbonate, polyethylene terephthalate, or polyester, and in some cases, an extracellular matrix protein hydrogel (collagen, fibrin…) is also added (Table 2).

Regarding to the membrane, its composition strongly influences barrier integrity and function, especially in terms of permeability, since cell–cell interactions might be conditioned [73,74]. Among the different parameters that are relevant to ensure an adequate contact among cells are pore diameter and density and membrane thickness [54]. In this line, Brown et al. developed a microfluidic device with a 0.2 μm pore polycarbonate (PC) membrane to establish a vascular and a brain chamber [62]. This model was integrated by endothelial cells, astrocytes, pericytes, and neurons embedded on collagen I to recreate a functional NVU and to evaluate barrier permeability (Figure 4A).

In addition to polycarbonate, the porous membrane might also be made of polyester, poly(D-L-lactic acid) (PDL-LA), or polyethylene terephthalate (PET) among other materials [50,55,56,57,58,59,60,61,62,63,66,67]. Each of them allows cells to grow; however, they might influence further assays, such as microscopy, due to changes in transparency [73]. The use of such porous membranes has further limitations, since they do not replicate the properties of BL; thus, its effect on barrier permeability is not considered [15]. Some researchers focused on developing porous membranes closer to BL. Singh et al. compared a Transwell membrane with crosslinked collagen and fibronectin, as model for the *in vivo* composition of basal lamina, to a traditional Transwell membrane and observed an improvement in barrier function [75]. Motallebnejad et al. took this concept further by developing a device with two chambers separated with a PET porous membrane and embedding astrocytes on hydrogel of collagen type I, Matrigel^®^, and hyaluronic acid [70]. Similarly, Santa-Maria et al. coated a polyester membrane with Matrigel^®^ to facilitate endothelial cells and bovine pericytes growth [58].

**Figure 4 bioengineering-10-00572-f004:**
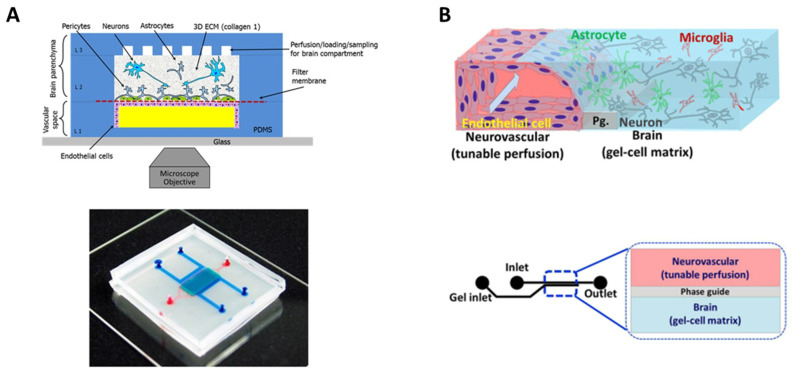
Biomimetic BBB-on-chip models compartmentalized by a membrane and hydrogel. (**A**) Schematic representation of the four-cell model developed by Brown et al. [62] with a polycarbonate membrane and a collagen I hydrogel (**top panel**), and a photograph of the assembled device loaded with colored dyes indicating different compartments: red = vasculature; semi-transparent white  =  filter membrane; turquoise = brain compartment; and blue = brain perfusion (**bottom panel**). OA publishing license: Adapted from Brown et al. [62]. (**B**) Schematic representation of the four-cell model developed by Koo et al. [76] with a collagen I hydrogel (**top panel**) and structure and format application of *in vitro* cultureware, OrganoPlate consisting of a 384-well microliter plate with two lanes (**bottom panel**). OA publishing license: Adapted from Koo et al. [76].

### 5.2. Hydrogels

Other authors explored the use of OOC devices that do not rely on semipermeable membranes. Koo et al. developed a microfluidic system in which cells from the cerebral chamber, namely astrocytes, microglia, and neuroblastoma cells, were embedded in a model of the extracellular matrix composed by collagen I [76]. Then, endothelial cells were seeded in the polymerized gel to mimic the vascular side (Figure 4B). In a later work, authors evaluated this model for rapid screening of potential neurotoxic compounds, pointing to the value of the model’s applicability [77]. Herland et al. also used collagen I to reproduce the extracellular matrix. In this case, authors developed a cylindrical collagen gel to mimic the *in vivo* geometrical disposition of the BBB [59]. The model integrated endothelial cells, astrocytes, and pericytes. Authors noticed that the presence of astrocytes and pericytes improved barrier function, thus confirming that the tri-culture better mimics *in vivo* conditions than endothelial cells in mono-culture.

In addition to collagen, hydrogels that replace a porous membrane might be made of other proteins, e.g., fibrin or combinations of proteins such as collagen a hyaluronic acid [59,76,77,78,79,80,81,82]. Hydrogel composition strongly influences cell growth, and thus must be carefully selected [83]. When it comes to developing *in vitro* models as realistic as possible, the ones lacking porous membranes are an interesting advance due to their allowing of a 3D disposition that might better mimic *in vivo* conditions [54,83].

## 6. BBB-on-Chip for Drug Evaluation

One of the fields in which BBB-on-chip models excel is drug testing. Currently, evaluation of drugs for the management of central nervous system-related disorders is a time-consuming, high-cost process, mainly due to the necessity of using animal models. The use of animal models, as previously discussed, might result in inaccurate data that lead to drug failure in clinical trials, and even increasing the time and costs needed to find novel drugs. Since microfluidic-based BBB *in vitro* models offer a biomimetic microenvironment and are cheaper than animal models, they are a strong alternative to reducing time and costs for drug evaluation. In this section, we will review the most recent BBB-on-chip models focused on drug testing, their strengths, and limitations.

These devices are of great interest for the evaluation of novel nanoparticles likely able to penetrate the BBB and reach the brain parenchyma. In this way, Hudecz et al. developed a microfluidic system containing endothelial cells and astrocytes separated with a silicon nitride membrane to evaluate a panel of nanoparticles [71]. Despite their model lacking the influence of pericytes and containing a porous membrane instead of hydrogel, authors found promising results regarding the influence of nanoparticle size and surface targeting moieties on the BBB penetration [22,54,83]. Therefore, BBB-on-chip models might be a useful tool for the designing of novel nanoparticles able to translocate across BBB. Ahn et al. developed a more complex model that consisted of a tri-culture of endothelial cells, pericytes, and astrocytes, which allowed for the quantification of a series of nanoparticle transports as well as mapping their distribution in the vascular and perivascular regions [60]. However, both chambers were separated with a polycarbonate membrane instead of with a hydrogel that would potentially better mimic *in vivo* conditions (Figure 5A).

In line with this, several models were proposed as novel platforms to evaluate nanoparticles and further drug effectivity. Some of those devices slightly improved the one proposed by Hudecz et al. by including pericytes as well as endothelial cells and astrocytes, namely Lee et al., Campisi et al., Noorani et al., and Sahtoe et al. [79,80,85,86]. However, devices developed by Lee et al. and Sahtoe et al. might be considered less biomimetic due to the presence of a semi-porous membrane instead of hydrogel [54,79,83,86]. Therefore, the device developed by Campisi et al. might be considered the most physiologically relevant of the discussed in this section since it included a tri-culture of endothelial cells, pericytes, and astrocytes, as well as a fibrin hydrogel to mimic the extracellular matrix [80]. However, it must also be carefully considered that a fibrin hydrogel might not really mimic *in vivo* extracellular matrix, and its replacement with further materials might result in an enhancement of the biomimetic consideration. In summary, the presence of these three different cell types, along with hydrogel instead of a porous membrane, might be considered as the minimum requisites to developing a biomimetic BBB *in vitro* microfluidic-based model.

Fengler et al. developed an *in vitro* BBB model using 3-lane OrganoPlates^®^ to develop a microfluidic system suitable for high-content compound screening [87]. This kind of device is of great interest for industrial and clinical purposes, since more than one drug candidate might be analyzed at once. The potential of their model, which consisted of induced pluripotent stem cells (iPSCs) differentiated into brain endothelial cells-like cells and pericytes, was evaluated with a library of anti-inflammatory compounds potentially able to penetrate the BBB. The authors concluded that their model was in fact able to discriminate between compounds able or unable to cross through the BBB and suggest that it might accelerate drug discovery for neurodegenerative disorder management. However, this model does not include a complex and functional NVU; thus, the influence of cells further than endothelial cells is not considered [22]. Wevers et al. developed a similar device based on 3-lane OrganoPlates^®^ as well that indeed fulfilled the two minimum requisites previously mentioned: their approach contained a tri-culture of endothelial cells, pericytes, and astrocytes embedded on a collagen I hydrogel (Figure 5B) [84]. Therefore, this BBB-on-chip model suitable for high-content compound screening might be considered as the most biomimetic to date.

Other authors included neurons in their BBB models to analyze the influence of those compounds able to cross through the barrier on the brain parenchyma. This is of the highest relevance in drug testing to identify experimental compounds that might produce neurotoxicity. In line with this, Vatine et al. developed a microfluidic system with endothelial cells on one chamber and astrocytes and neurons on the other. Both chambers were separated with a porous poly(dimethyl)siloxane membrane [88]. These device results are interesting due to the presence of neurons, since the potential neurotoxicity of a certain drug might be evaluated. However, as with previously discussed models, this lacks pericytes and the porous membrane should be replaced with hydrogel to enhance biomimetic properties [22,54,83]. Maoz et al. coupled two chips, one containing a tri-culture of endothelial cells, pericytes, and astrocytes, and the other one containing astrocytes and neurons to reproduce brain parenchyma [89]. Medium was firstly flowed through the BBB-on-chip and then fluid effluent was transferred to the second chip. Authors evaluated the effect of methamphetamine on both the BBB and the brain parenchyma and observed the reversible effect previously noticed during *in vivo* experiments. Therefore, this model might be of interest to evaluate potential neurotoxicity *in vitro.*

BBB-on-chip systems offer further interesting advantages for drug evaluation, such as investigating the influence of hormones on drug transport through the BBB. Brown et al. developed a microfluidic system to determine whether cortisol influences opioid transport to the brain parenchyma [90]. The device consisted of iPSCs differentiated into brain endothelial cells in one chamber and astrocytes in the other. Both chambers were separated by a polyethylene terephthalate membrane. The authors observed that cortisol can indeed modify opioid transport through the BBB by a direct interaction with endothelial cells. These preliminary data are of great interest; however, the model might be improved by including pericytes and by replacing the porous membrane with hydrogel [22,54,83].

Yu et al. developed a microfluidic system to mimic neuroinflammation as a platform for the evaluation of neuroinflammatory compounds [78]. The device consisted of endothelial cells, pericytes, and astrocytes isolated from rat embedded in collagen I hydrogel. Tumor necrosis factor alpha was added to reproduce neuroinflammation, and dexamethasone to mitigate it. This proposal is of great interest due to the tri-culture and the use of hydrogel to mimic *in vivo* tissue microarchitecture. However, one of the limitations of this device lies in the use of rat-derived cells, since some authors noticed a different behavior compared to human cells, e.g., regarding P-gp expression levels [28]. Microfluidic devices focused on neuroinflammation analysis might benefit from containing immunosensors such as the one proposed by Su et al., able to detect three different cytokines relevant to this process [91]. The BBB-on-chip model developed in this work, however, displayed several issues regarding its biomimetic profile, since it contains mouse brain endothelial cells on a silicon-nitride membrane. Matthiesen et al. developed a microfluidic system that also was coupled with biosensors [92]. In this case, authors aimed to follow barrier alterations in response to nitrosative stress and the potential capacity of the antioxidant N-acetylcysteine to reverse damage. The proposed model consisted of endothelial cells and astrocytes separated by a polycarbonate membrane; and thus, might not be considered realistic enough. Nevertheless, this approach is a promising starting point for the development of microfluidic systems that allow continuous monitoring of different parameters. To date, one of the most accurate models for reproducing neuroinflammation *in vitro* is the one developed by Herland et al., mentioned in previous sections of the present review [59]. In this case, authors mimicked neuroinflammation with TNF-α and a quantified cytokine release profile to investigate the role of the BBB on the process.

The BBB-on-chip models might also be helpful for investigating strategies to enhance barrier permeability to allow impermeable drugs to reach brain parenchyma. As an example, Bonakdar et al. developed a microfluidic device consisting of endothelial cells seeded on a polyester membrane to separate two chambers to evaluate the influence of pulsed electric fields [65]. Despite the fact that this model might be considered very simple in terms of cell composition and the presence of a biomimetic extracellular membrane, it is an interesting first step to reduce the necessity of animal models to study the influence of pulsed electric fields on drug permeability.

Organ-on-chip technology offers the possibility of combining more than one *in vitro* model to achieve results even closer to *in vivo* observations. In line with this, Koenig et al. developed a microfluidic platform containing 3D *in vitro* liver, neural, and BBB models in order to evaluate the permeability of propranolol and atenolol to brain tissue upon drug metabolization in the liver (Figure 6). Authors found the permeation behavior of both drugs significantly close to the one observed *in vivo*; however, as pointed out in the study, their BBB model did not include pericytes and astrocytes [22]. Moreover, endothelial cells were seeded on a polyester membrane instead of on hydrogel that mimics better basal lamina [54,83]. Therefore, these results should be considered promising, but might be improved. Authors mentioned that, in future steps, they plan improving the BBB model, as well as including intestinal and renal *in vitro* models to obtain more accurate data.

## 7. Mimicking Brain Pathology to Evaluate Potential Treatments

BBB-on-chip technology might be helpful for the evaluation of drug potential to treat specific diseases that affect the brain parenchyma (Figure 7). These devices might be useful to evaluate the capacity of a compound or nanoparticle to penetrate the BBB and reach brain tissue. Moreover, including neurons in the *in vitro* model might help in evaluating drug neurotoxicity. Finally, microfluidic chips allow for mimicking certain pathologies that trigger BBB disruption.

### 7.1. Brain Cancer

Peng et al. developed an interesting microfluidic device focused on the evaluation of potential drugs for glioblastoma management [94]. The microfluidic channels of this device were modified in situ with a photocrosslinkable copolymer, which further ensured a stable and evenly distributed coating with extracellular matrix proteins. The BBB model consisted of endothelial cells on the ‘vascular’ chamber and pericytes and astrocytes on the ‘brain’ chamber. Later, U87 cells were embedded in Matrigel^®^ to establish an *in vitro* 3D glioblastoma model that was seeded on the brain chamber (Figure 8A). Finally, the authors evaluated the capacity of a series of nanoparticles to cross through the BBB and penetrate within glioblastoma cells. Tricinci et al. developed a different device with the same purpose. In this case, authors used endothelial cells and astrocytes to develop the BBB model and developed U87 spheroids loaded in microcages that were implanted in the ‘brain’ chamber of the chip. Then, the ability of lipid nanocarriers to cross through the BBB model and to reduce tumor cell viability was analyzed. Shi et al. developed an *in vitro* 3D glioma model (U251 cells embedded in Matrigel^®^) that was coupled with a tri-culture of endothelial cells, pericytes, and astrocytes to mimic the BBB [95]. Vascular and brain chambers were separated with a polycarbonate membrane coated with Matrigel^®^ to facilitate cell adhesion. The resultant microfluidic device was used to evaluate the potential of a panel of traditional Chinese medicine compounds towards glioma (Figure 8B).

Li et al. introduced an interesting modification in their BBB–glioma model, similar to that previously discussed by Koenig et al. [93,96]. Given that hepatic metabolism of anti-tumor drugs for glioma management might influence drug bioavailability, authors developed a microfluidic system that integrated liver, BBB, and glioma models. The chip consisted of three separated channels. HepG2 cells, as a model of liver tissue, and U87 cells, as a model of glioma, were grown on separated channels, whereas endothelial cells and astrocytes embedded in collagen were included on the third channel. Then, the authors evaluated the influence of liver metabolism on the ability of three anticancer drugs to penetrate the BBB and decrease U87 cell viability. Despite these innovations, the BBB model lacked other relevant cells, such as pericytes. Overall, the proposed work showed the advantages of microfluidic-based models for traditional drug evaluation methods.

### 7.2. Alzheimer’s Disease

The development of novel therapies for the management of Alzheimer’s disease (AD) might also be improved using BBB-on-chip models. Wang et al. synthetized a nanoplatform able to inhibit amyloid-beta aggregation upon laser irradiation [97]. To evaluate its capacity to penetrate the BBB, authors developed a microfluidic model of the BBB consisting of two chambers separated with a polyester membrane. Endothelial cells were grown on one chamber and astrocytes on the other one (Figure 9A). Then, neuroinflammation was mimicked with TNF-α and the nanoplatform was added to the vascular channel. Authors observed that the novel nanomedicine was indeed able to penetrate the *in vitro* barrier, thus suggesting promising *in vivo* results. Other authors focused on developing more accurate *in vitro* models of AD that include the BBB to investigate novel drugs. In this line, Shin et al. developed an *in vitro* AD model consisting of neural progenitor cells with a mutation in the APP gene that leads to the extracellular deposition of amyloid plaques [98]. Those cells were grown on a Matrigel^®^ culture and seeded on a microfluidic chip with endothelial cells to mimic the BBB. Both chambers were separated with hydrogel to allow for close contact among both cell types (Figure 9B). Authors observed that barrier permeability was increased in the presence of the AD model in comparison with cells without the mutation in the APP gene. This might lead to the penetration of neurotoxic compounds that might damage neurons, according with further results. The authors also found that the reduction in amyloid-beta generation resulted in an amelioration of BBB function. Therefore, authors concluded that this device might be of interest for evaluating novel drugs for AD. Combining this AD model with a more accurate BBB model that includes pericytes and astrocytes might be interesting to better reproduce *in vivo* tissue microarchitecture.

### 7.3. Virus Infection

Virus infection and its management might also be studied using BBB-on-chip models. Boghdeh et al. developed a microfluidic system with a tri-culture of endothelial cells, pericytes, and astrocytes separated with a polyethylene membrane and a collagen and fibronectin mix to mimic extracellular matrix [100]. This device was exposed to the Venezuelan equine encephalitis virus and further to the drug omaveloxolone. Authors observed that pre-treatment with this drug ameliorated virus-induced damage. Therefore, authors concluded that their model might be useful to accelerate the development of novel drugs towards this pathogen.

BBB-on-chip models might be helpful to understand and ameliorate the negative impact of SARS-CoV-2 infection on the BBB. Buzhdygan et al. reported that SARS-CoV-2 subunit S1 increased BBB permeability on a microfluidic model consisting of endothelial cells embedded on hydrogel of collagen I, hyaluronan, and Matrigel^®^ [101]. The same microfluidic model was further used for an in-depth study of the effect of virus infection on BBB integrity. DeOre et al. found that viral spike protein activated RhoA, thus triggering vascular integrity disruption [102]. With this in mind, Suprewicz et al. developed a novel microfluidic system with hydrogel with cylindrical voids in order to reproduce *in vivo* geometry, and endothelial cells were grown to mimic the BBB [103]. Authors found that recombinant human plasma gelsolin was able to ameliorate barrier dysfunction triggered by spike protein. Authors suggest that these data might be extrapolated to clinical practice for the management of cases of severe COVID-19 infection. However, it must be considered that these models do not include pericytes, astrocytes, or neurons, thus the potential neurotoxicity of plasma gelsolin was not evaluated.

### 7.4. Ischemia Stroke

Wevers et al. developed a BBB-on-chip model and a protocol to mimic ischemia stroke suitable for high throughput evaluation of drugs for restorative therapies [99]. Endothelial cells, astrocytes, and neurons were grown on a 3-lane OrganoPlate^®^ microfluidic system and were exposed to chemical hypoxia, hypoglycemia, and halted perfusion to simulate the damage induced by ischemic stroke (Figure 9C). Authors were able to reproduce ischemic stroke *in vivo*, although the influence of pericytes is not considered in their model. Lyu et al. were a step further on the designing of their *in vitro* model of ischemic stroke [104]. Firstly, their device contained each cell type relevant for a functional NVU. Astrocytes, neurons, and microglia were embedded into hydrogel and placed on the brain chamber, whereas endothelial cells and pericytes were grown on the blood side of the chip. Then, authors mimicked ischemic stroke and evaluated the potential of stem cells, which were injected into the blood side channel to restore the damage. The proposed model meets all the required features to be considered a robust, biomimetic *in vitro* model suitable for the evaluation of neurorestorative therapies for ischemic stroke.

## 8. Conclusions and Future Perspectives

In previous sections of the present review, different BBB-on-chip models were discussed, as well as their potential application for drug evaluation. This review allowed for the establishment of the two minimum requisites that should fulfill a microfluidic model to be considered biomimetic: The first one is a tri-culture of human endothelial cells, astrocytes, and pericytes, since the three of them are responsible for the unique properties of the BBB. Animal-derived cell lines should not be considered due to the differences between human cells in terms of protein expression and other features. Secondly, traditional semipermeable membranes should be replaced by biological membranes based on extracellular matrix proteins or hydrogels mimicking basal lamina to ensure close cell–cell contact. Furthermore, microfluidic models display the intrinsic advantage of considering the influence of shear stress due to a continuous fluid flow, a key parameter for brain endothelial cells.

Moreover, it must be carefully considered that, when characterizing the utility of the BBB-on-chip model, TEER measurement by itself might not be an adequate parameter to ensure model validity. Further parameters, such as reference substances permeability or expression of key proteins, must be analyzed to guarantee that the model is biomimetic enough to translate the obtained results to the clinic.

Based on these considerations, herein we also mentioned certain improvements for developing next-generation BBB-on-chip models for drug evaluation, given that OOC technology allows for the development of a wide range of microfluidic devices according to the different experimental purposes. The material used to fabricate the devices should be carefully selected, however, since the intrinsic physicochemical properties of some of them might limit their use. For example, polydimethylsiloxane absorbs a wide range of biochemicals, thus is inadequate for drug screening assays. Thermoplastics such as polystyrene, poly(methyl methacrylate), polycarbonate, or cyclic olefin copolymer display low surface absorption and might be more interesting materials for developing microfluidic devices for small-molecule screening [105].

Some authors noticed that mimicking *in vivo* geometry in chip models might enhance barrier integrity, thus providing more robust results in terms of drug candidates’ permeability. Including neurons in the *in vitro* models might be of the highest relevance to evaluating the potential neurotoxicity of novel drug candidates. Moreover, OOC technology allows the development of multi-organ microfluidic systems to consider drug absorption and metabolism prior to the BBB entrance and its later excretion. Such improvements make BBB-on-chip models increasingly versatile alternatives to animal models in preclinic phases of drug development.

## Figures and Tables

**Figure 1 bioengineering-10-00572-f001:**
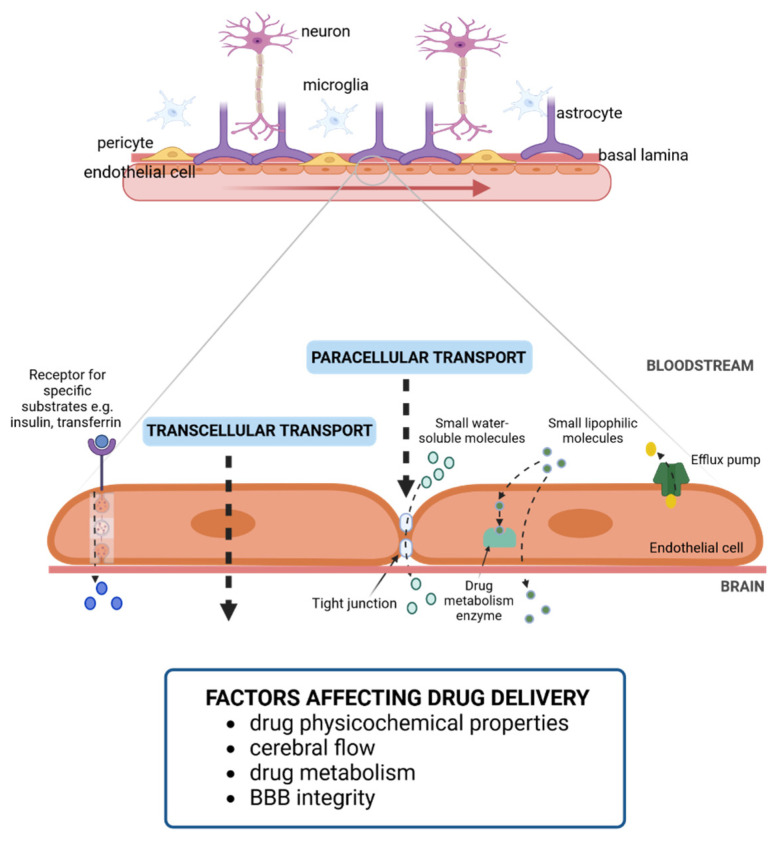
Schematic representation of the blood–brain barrier and the transport pathways across the endothelial cells. The BBB is composed of endothelial cells, pericytes, astrocytic foot processes, neurons and microglia, as well as the basal lamina. Paracellular and transcellular transport across endothelial cells allow for the drug delivery to CNS. The main factors that influence drug delivery are also included. Created with BioRender.com (accessed on 4 May 2023).

**Figure 2 bioengineering-10-00572-f002:**
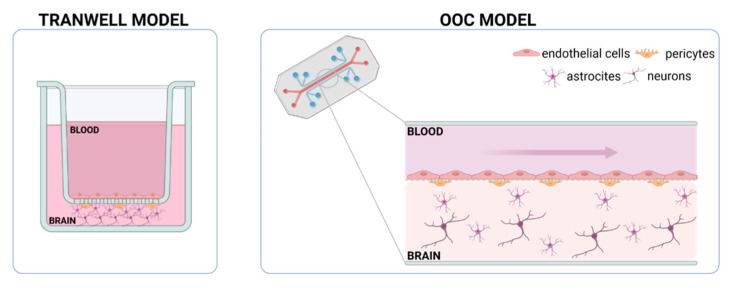
Schematic representation of an *in vitro* BBB model based on a Transwell insert (static conditions) and on an OOC microfluidic device (flow conditions). Both models present two separate compartments, reproducing the vascular zone (BLOOD) and the nervous tissue (BRAIN). To recreate the architecture of the BBB and compartmentalize the cell culture, a polycarbonate membrane is used in Transwell models, while other types of membranes and/or hydrogels can be used in the microdevices. In the proposed models, the co-cultures of the main cell types present in the BBB (endothelial cells, pericytes, and astrocytes), as well as the neurons of the brain tissue, were integrated.

**Figure 3 bioengineering-10-00572-f003:**
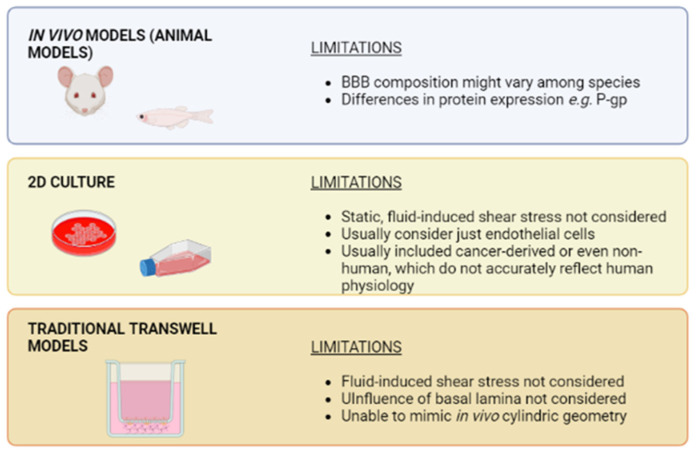
Limitations of traditional *in vitro* models of the blood–brain barrier, from animal models to traditional cell culture systems. Created with BioRender.com (accessed on 27 April 2023).

**Figure 5 bioengineering-10-00572-f005:**
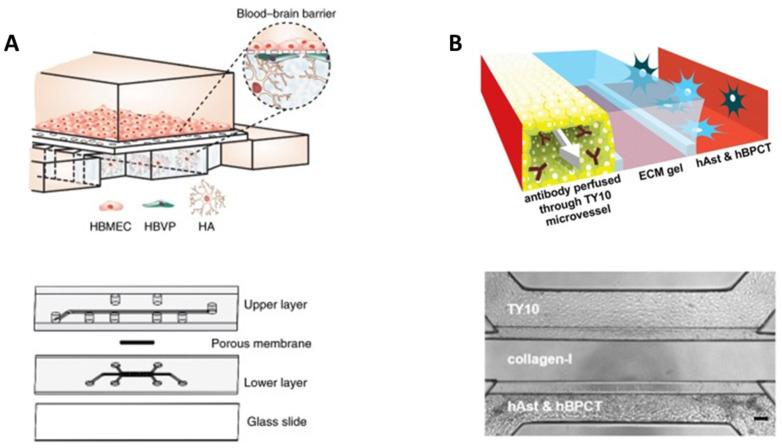
Microfluidic models developed for testing nanoparticle permeability across the BBB. (**A**) Schematic representation of the three-cell model from Ahn et al. [60] with a porous polycarbonate membrane (**top panel**) and explosion view of the device consisting of an upper vascular layer, porous membrane, lower perivascular layer, and glass slide (**bottom panel**). OA publishing license: Adapted from Ahn et al. [60]. (**B**) 3D artist impression of the center of the three-cell model developed by Wevers et al. [84] (**left panel**) and phase contrast images of the BBB co-culture on day 7, scale bar 100 µm (**right panel**). OA publishing license: Adapted from Wevers et al. [84].

**Figure 6 bioengineering-10-00572-f006:**
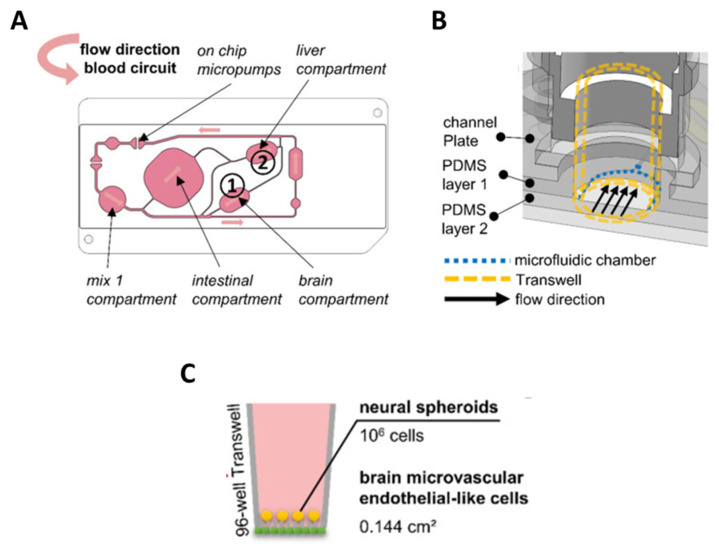
Multi-organ microfluidic system developed by Koenig et al. [93]. (**A**) 2D view of the microfluidic chip; the surrogate blood circuit is shown in pink. In the depicted circuit, a medium reservoir (mix 1) is interconnected with a 24-well intestinal compartment and 96-well compartments for the liver ① and BBB/brain ② model. The fluid flow is created by on-chip micropumps, and the direction of the flow is indicated by arrows. (**B**) 3D view of the brain culture compartment. The bottom of the compartment consists of the PDMS layer 2. At the sides, the compartment consists of the PDMS layer 1 and the channel plate. Cut-off 96-well Transwells (yellow dotted line) can be inserted into the compartment and stand at their edges on a 100 µm-high step of PDMS layer 2. Endothelial cells cultured at the bottom of the Transwell membrane are thereby directly exposed to the fluid flow passing underneath. (**C**) Schematic representation of the brain microvascular endothelial-like cells and neural spheroids were combined in 96-well Transwells to build a blood–brain-barrier/brain model. OA publishing license: Adapted from Koenig et al. [93].

**Figure 7 bioengineering-10-00572-f007:**
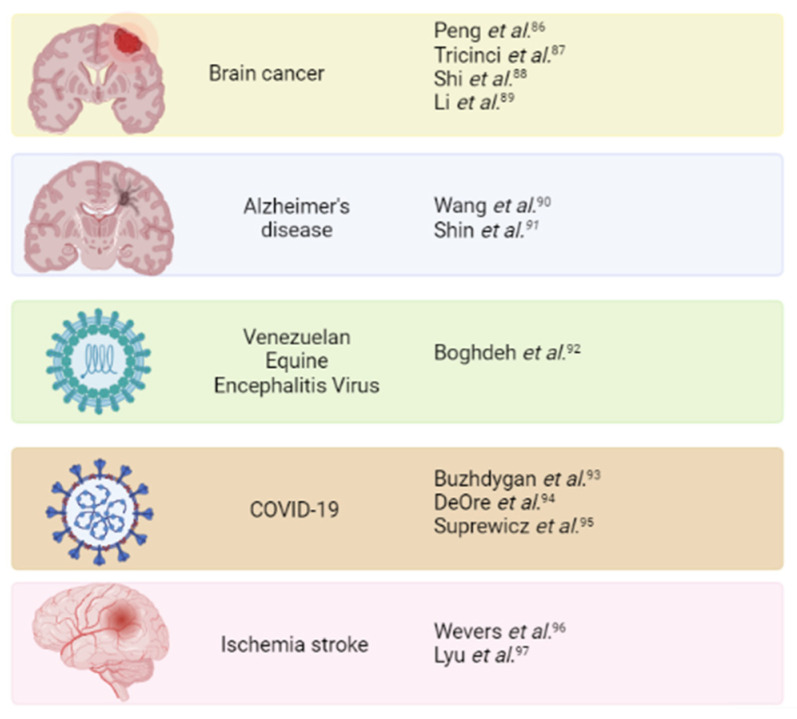
Compilation of BBB-on-chip models for the evaluation of novel drugs for brain-related pathologies management. With BioRender.com (accessed on 14 February 2023).

**Figure 8 bioengineering-10-00572-f008:**
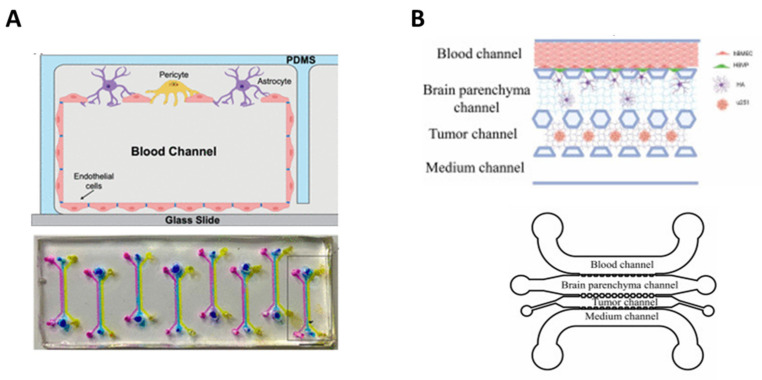
BBB-on-chip models for the evaluation of anti-glioblastoma drug candidates. (**A**) Three-cell model developed by Peng et al. [94] (**top panel**) and top view of the chip, composed of eight independent μBBB units with each unit having three main channels, scale bar: 500 μm (**bottom panel**). Adapted with permission from Peng et al. [94]. Copyright © 2020, American Chemical Society. (**B**) Schematic illustration of the BBB-U251 chip and 3D culture of U251 by Shi et al. [95]. Adapted with permission from Shi et al. [95]. Copyright © 2022, Elsevier B.V.

**Figure 9 bioengineering-10-00572-f009:**
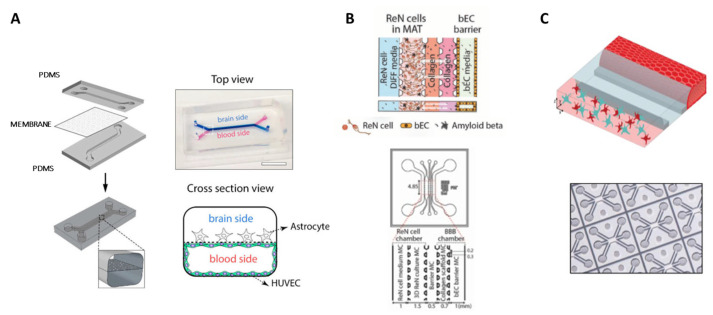
BBB-on-chip models to evaluate Alzheimer’s disease drug candidates and ischemia stroke. (**A**) Schematic representation of the model developed by Wang et al. [97] (**top panel**) and the top view of the chip filled with blue ink and red ink for the brain side and blood side, respectively, scale bar 10 mm (**bottom panel**). OA publishing license: Adapted from Wang et al. [97]. (**B**) Model developed by Shin et al. [98] (**top panel**) and detail of the measurement of each chamber of the chip (bottom panel). OA publishing license: Adapted from Shin et al. [98]. (**C**) 3D artist impression of the three-cell model developed by Wevers et al. [99] (**top panel**) and picture of the bottom of the OrganoPlate, showing several three-lane chips (**bottom panel**). OA publishing license: Adapted from Wevers et al. [99].

**Table 1 bioengineering-10-00572-t001:** Role of the cellular components of the neurovascular unit (NVU).

Cell Type	Function
Endothelial cell	Responsible for high selective permeability of the BBB
	Production of BL components
Pericytes	Maintenance of tight junctions and barrier function
	Production of BL components
	Regulation of vesicle endothelial transcytosis
	Regulation of immune cells infiltration
Astrocytes	Endothelial cells maintenance
	Connection of neurons with endothelial cells
	Regulation of immune cells infiltration
Microglia	Phagocytosis Immune response activation Maintenance of BBB integrity by close connection with pericytes
Neurons	Indirect control of blood flow
	Promotion of angiogenesis during the development

**Table 2 bioengineering-10-00572-t002:** Substrates present in microfluidic devices to achieve the BBB cellular architecture. In the literature, porous membranes based on inert material and hydrogels can be used to mimic the extracellular matrix.

Semipermeable Porous Membrane	Hydrogel	References
Polycarbonate	Collagen and hyaluronic acid	Ahn et al. [60] Boot et al. [61] Brown et al. [62] Jeong et al. [55] Shao et al. [63] Wang et al. [64]
Polyester	Collagen	Bonakdar et al. [65] Choi et al. [66] Falanga et al. [67] Santa-Maria et al. [58]
Poly(D-L-lactic acid)	Fibrin	Mancinelli et al. [68]
Polyethylene terephthalate	-	Meena et al. [69] Motallebnejad et al. [70] Tu et al. [56]
Silicon	-	Hudecz et al. [71] Mossu et al. [72]

## Data Availability

Not applicable.

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
