# Peer review of "Towards Novel Biomimetic In Vitro Models of the Blood–Brain Barrier for Drug Permeability Evaluation"

_bioengineering, 2023, doi:10.3390/bioengineering10050572_

Round 1

Reviewer 1 Report

My comments are as follows:

1. The title of the manuscript should be modified.

2. Similar studies already reported related to the proposed title, therefore, the contents of the manuscript should be specified.

3. Section no2: "Challenges in drug delivery to the CNS" looks general explanation that should be more emphasized.

4. Section no4.  The mechanism of biomimetic models of the BBB should be provided.

Moderate editing of the English language is required throughout the text. 

Reviewer 2 Report

-The aims and outline of the paper must be introduced.

-Line 33, the sentence "Regarding BBB composition, a cellular and a non-cellular component can be distinguished" is not clear and must be restated.

-I suggest to represent the functions of the different cell types of neurovascular unit in a Figure.

- More descriptions about the function of neurons must be added.

- It must be described how paracellular transport works.

-The full name of LDL particles must be mentioned.

-A diagram demonstrating the barriers for the delivery to CNS and the effect of factors affecting the successful delivery must be added.

-For biomimetic models of the BBB, firstly, important components to include, must be addressed clearly.

 Secondly, it must be addressed if any traditional model other than transwell has already been used.

-A diagram must be added for transwell model.

- The general components of an OOC device must be shown clearly and a scheme/figure must be added.

-Subsections must be added to address different types of OOC device.

-Subsections must be added for drug evaluation/potential treatments based on drug type, treatment or disease.

-Some directions for the future generation of OOC devices must be added.

Reviewer 3 Report

The current animal and in vitro models for brain-related pathologies and drug evaluation have limitations that result in preclinical drug candidates failing in clinical trials due to their inability to mimic the blood-brain barrier (BBB). The article reviews recent advances in organ-on-chip models of the BBB, which provide a promising alternative to traditional models. These microfluidic models recreate the architecture of the BBB and mimic the fluidic conditions of the cerebral microvasculature, potentially providing robust and reliable data on drug candidate ability to reach the brain parenchyma. The article is well-organized and informative, and I recommend it for publication in Bioengineering with minor revision

1. In Fig. 2, the animal model could be an example of an in vivo model.

2. Additional information could be included in Fig. 2, such as the fact that traditional in vitro models often use non-human or cancer-derived cell lines that do not accurately reflect human physiology and disease conditions. Another limitation is that traditional in vitro models are static and cannot account for the dynamic changes in BBB function that occur in response to physiological or pathological stimuli, such as inflammation or drug exposure.

Round 2

Reviewer 2 Report

The comments are applied.

Quality of English language is acceptable.